# Transcription Factors in Alkaloid Engineering

**DOI:** 10.3390/biom11111719

**Published:** 2021-11-18

**Authors:** Yasuyuki Yamada, Fumihiko Sato

**Affiliations:** 1Laboratory of Medicinal Cell Biology, Kobe Pharmaceutical University, Kobe 658-8558, Japan; 2Department of Plant Gene and Totipotency, Division of Integrated Life Science, Graduate School of Biostudies, Kyoto University, Kyoto 606-8502, Japan; 3Graduate School of Science, Osaka Prefecture University, Sakai 599-8531, Japan

**Keywords:** benzylisoquinoline alkaloid, monoterpene indole alkaloid, nicotine, steroidal glycoalkaloid, transcription factor, jasmonate signaling, gene cluster, metabolic engineering, transcriptional network

## Abstract

Plants produce a large variety of low-molecular-weight and specialized secondary compounds. Among them, nitrogen-containing alkaloids are the most biologically active and are often used in the pharmaceutical industry. Although alkaloid chemistry has been intensively investigated, characterization of alkaloid biosynthesis, including biosynthetic enzyme genes and their regulation, especially the transcription factors involved, has been relatively delayed, since only a limited number of plant species produce these specific types of alkaloids in a tissue/cell-specific or developmental-specific manner. Recent advances in molecular biology technologies, such as RNA sequencing, co-expression analysis of transcripts and metabolites, and functional characterization of genes using recombinant technology and cutting-edge technology for metabolite identification, have enabled a more detailed characterization of alkaloid pathways. Thus, transcriptional regulation of alkaloid biosynthesis by transcription factors, such as basic helix–loop–helix (bHLH), APETALA2/ethylene-responsive factor (AP2/ERF), and WRKY, is well elucidated. In addition, jasmonate signaling, an important cue in alkaloid biosynthesis, and its cascade, interaction of transcription factors, and post-transcriptional regulation are also characterized and show cell/tissue-specific or developmental regulation. Furthermore, current sequencing technology provides more information on the genome structure of alkaloid-producing plants with large and complex genomes, for genome-wide characterization. Based on the latest information, we discuss the application of transcription factors in alkaloid engineering.

## 1. Introduction

Plants produce structurally divergent low-molecular-weight secondary metabolites (recently termed as “specialized metabolites”) to protect their bodies from the attack of pathogens and herbivores, and/or to interact with beneficial organisms, such as to attract symbiotic microorganisms and pollinators. Among secondary metabolites, nitrogen-containing alkaloids found in a limited number of plant species are often more biologically active and are thus used as pharmaceuticals [1]. Because of the diversity of the chemical structures and biosynthetic/evolutional origins of such compounds, the elucidation of their biosynthetic pathways and regulatory mechanisms of biosynthesis has been limited to a few plant species. In this review, we focus on transcription factors (TFs) that regulate alkaloid biosynthesis in comparison with those of more universal secondary metabolites, such as phenylpropanoids and terpenoids and their potential use in alkaloid engineering.

## 2. General Characteristics of TFs

TFs regulate gene expression involved in numerous plant processes through specifically binding to *cis*-acting elements in the promoters of target genes. Several TFs involved in the regulation of alkaloid biosynthesis have been characterized, including those of nicotine alkaloids in *Nicotiana* species; steroidal glycoalkaloids (SGAs) such as tomatine in *Solanum lycopersicum* (Solanaceae) and solanine in *Solanum tuberosum* (Solanaceae); monoterpenoid indole alkaloids (MIAs) such as vinblastine and vincristine in *Catharanthus roseus* (Apocynaceae); and benzylisoquinoline alkaloids (BIAs) such as berberine in *Coptis japonica* (Ranunculaceae), sanguinarine in *Eschscholzia californica* (Papaveraceae), and morphine in *Papaver somniferum* (Papaveraceae) [2,3]. Detailed TFs reported in the alkaloid pathways are listed in Table 1 and described below.

### 2.1. APETALA2/Ethylene-Responsive Factor (AP2/ERF) TFs

AP2/ERF family proteins form one of the largest groups of plant-specific TFs. The members contain an AP2/ERF domain that consists of approximately 60 amino acid residues and is involved in DNA-binding. Based on the number of AP2/ERF domains, the AP2/ERF superfamily can be divided into three subfamilies: AP2 (two AP2/ERF domains), RAV (one B3 domain in addition to one AP2/ERF domain), and ERF (one AP2/ERF domain) [44]. The ERF family can be further divided into the dehydration-responsive element binding (DREB) protein subfamily, whose members are well known to be involved in drought tolerance, and the ERF subfamily, whose members are involved in various stress responses. The DREB and ERF subfamily proteins are known to bind to the dehydration-responsive element (e.g., GCCGAC) and the GCC box (e.g., AGCCGCC), respectively [45]. The AP2 and RAV family proteins are involved in plant growth and flower development, whereas the DREB and ERF subfamily proteins play an important role in biotic and/or abiotic stress responses, disease resistance, and plant hormone signaling. Jasmonate (JA)-responsive members of the ERF subfamily, which predominantly belong to the Group IX subgroup, play a key role in the biosynthesis of plant-specialized metabolites, including those of several alkaloids [3,17,18,46,47]. Group IX *ERF* genes frequently form gene clusters in nicotine, SGA, and MIA biosynthesis, which suggests the evolutionary significance of *ERF* clusters [48].

### 2.2. WRKY TFs

WRKY superfamily proteins are also found mainly in plants. Many WRKY TFs have been shown to play pivotal roles in plant development, senescence, and defense responses [49,50]. All the WRKY proteins contain at least one WRKY domain composed of approximately 60 amino acid residues, which includes a highly conserved WRKYGQK sequence at the N-terminal and a zinc finger motif at the C-terminal [51]. The WRKY superfamily can be divided into three groups: Group I (two WRKY domains with a C2H2-type zinc finger), Group II (one WRKY domain with a C2H2-type zinc finger), and Group III (one WRKY domain with a C2HC-type zinc finger). Group II WRKYs can be further classified into five subgroups: IIa, IIb, IIc, IId, and IIe. The WRKY domain typically binds to the W-box DNA sequence motif (e.g., TTGACT) [50]. Recent structural analyses revealed that the conserved WRKYGQK motif with a beta-sheet structure binds to the major groove of the DNA strand [52,53]. Several JA-responsive WRKY TF proteins belonging to different groups have been reported to be involved in the regulation of alkaloid biosynthesis, as well as that of other specialized metabolites [2,3,22,23,24,54].

### 2.3. Basic Helix–Loop–Helix (bHLH) TFs

bHLH superfamily proteins are broadly distributed in eukaryotic kingdoms and are involved in many essential biological processes. In animals, bHLH TFs can be classified into six groups (A to F) that regulate circadian rhythm, cell cycle, and developmental processes [55,56]. In plants, bHLH proteins were first divided into 12 groups (Groups I to XII) based on the phylogenetic tree of *A. thaliana* bHLH TFs [57]. However, after sequencing the genomes of several plant and algal species, bHLH TFs were further classified into 32 subfamilies by means of phylogenetic analysis [58]. Plant bHLH TFs are involved in a wide array of physiological and developmental processes, including light and phytohormone signaling, biotic/abiotic stress responses, and tissue and organ development [59,60]. In addition, bHLH TFs have also been shown to regulate the biosynthesis of plant-specialized metabolites, including alkaloids [2,3,46,61,62]. The bHLH domain consists of approximately 60 amino acids, with two functionally different regions. The N-terminal end of the bHLH domain, which is composed of 15–20 amino acid residues in the basic region, is involved in DNA-binding, while two amphipathic alpha helices with a linking loop of variable lengths and sequences in the HLH region are involved in the formation of homodimeric or heterodimeric complexes. Based on their DNA-binding properties [55], most bHLH proteins in plants belong to Group B and specifically bind to the E-box (e.g., CANNTG).

MYC2, found in JA-insensitive *jai1*/*jin1* mutants of *A. thaliana*, is a key component of the JA signaling pathway [63,64]. JA promotes the degradation of jasmonate ZIM domain (JAZ) repressor proteins by CORONATINE-INSENSITIVE 1 (COI1) and 26S proteasome and the release of MYC2 for the upregulation of JA-responsive genes. MYC2-type bHLH TFs also play an essential role in the regulation of specialized metabolism in many plant species [3,65,66,67,68], whereas non-MYC2-type bHLH TFs have been reported to control the biosynthesis of BIA and might be a type of bHLH protein in BIA-producing plant species [69]. In addition, subgroup IV bHLH TFs based on the phylogenetic analysis of AtbHLH proteins have been shown to regulate triterpene saponin and MIA biosynthesis in several plant species [31,70,71]. These JA-responsive MYC2-type and non-MYC2-type bHLH TFs orchestrate transcriptional changes in the alkaloid biosynthetic pathways.

### 2.4. Other TFs

MYB superfamily proteins are widely distributed in plants and play important roles in plant growth, development, and stress responses. MYB proteins contain an MYB DNA-binding domain composed of approximately 50 amino acid residues. The MYB domain generally comprises one to four imperfect repeats. Plant MYB TFs can be classified into four groups: R2R3-MYB, R1R2R3-MYB, 4R-MYB, and MYB-related proteins. Although many R2R3-MYB TFs have been reported to be involved in the regulation of specialized metabolites, such as glucosinolate, camalexin, and in flavonoid biosynthesis [3,72], only a few MYB TFs involved in the regulation of alkaloid biosynthesis have been reported thus far [37,38,39].

The basic region/leucine zipper (bZIP) superfamily is found in all eukaryotes and plays a critical role in defense and stress responses, phytohormone signaling, and senescence in plants. The bZIP proteins are typically divided into 10 groups (A–I and S) in *A. thaliana*. Two bZIP TFs have been identified in *C. roseus* and confirmed to bind to a G-box-like sequence. In addition, C2H2-type zinc finger proteins, GATA TF, and AT-hook motif nuclear localized TFs have also been reported.

## 3. TFs Involved in the Biosynthesis of Universal Secondary Metabolites, Such as Phenylpropanoids and Terpenoids, in the Model Plant, *Arabidopsis*

Many research groups have elucidated the regulatory mechanisms of phenylpropanoid and terpenoid biosynthesis, which are commonly found in almost all plant species. In the present study, we briefly describe several TFs that are involved in the regulation of phenylpropanoid, terpenoid, and camalexin biosynthesis in *A. thaliana*.

### 3.1. TFs in the Phenylpropanoid Pathway

In the biosynthesis of phenylpropanoids including anthocyanin (flower pigments), the complex that includes WD40-repeat (WD40) protein, Transparent Testa Glabra 1 (TTG1), R2R3-MYB TFs such as Production of Anthocyanin Pigment 1 (PAP1)/MYB75, PAP2/MYB90, MYB113, and MYB114, and bHLH TFs such as Glabra 3 (GL3), Enhancer of GL3 (EGL3), and Transparent Testa 8 (TT8) acts as an important regulatory module and regulates the expression of the “late” anthocyanin biosynthetic enzyme genes in *A. thaliana* [61,73]. This MYB–bHLH–WD40 (MBW) complex is widely distributed in phenylpropanoid biosynthesis [74] (Figure 1A).

Developmental signals and environmental stresses, such as sugar, UV light, temperature, and hormone signaling, are known to control anthocyanin biosynthesis. For instance, sucrose-induced anthocyanin biosynthesis depends on the function of PAP1/MYB75 [75]. Gibberellin (GA) represses sucrose-induced anthocyanin production because it induces the degradation of DELLA proteins, which are negative regulators of GA signaling, and the interaction of MYB-LIKE2 (MYBL2) with TT8, resulting in the suppression of anthocyanin biosynthesis [76,77]. JA is also involved in sucrose-induced anthocyanin biosynthesis, mediated by the interaction of JAZ proteins with the MBW complex [76,78,79], suggesting that the COI1–JAZ receptor complex may be conserved in many plant species and involved in the biosynthesis of various secondary metabolites, as discussed below. Ethylene (ET) inhibits sugar- and light-induced anthocyanin production [80]. ETHYLENE-INSENSITIVE 3 (EIN3), a key factor in ET signaling, directly suppresses the expression of TT8 [81]. High light, UV light, or cold stress also induces anthocyanin biosynthesis via the MBW complex or ELONGATED HYPOCOTYL 5 (HY5) TF [82,83,84,85]. A brief model of signal transduction involved in anthocyanin biosynthesis is described in Figure 1A.

### 3.2. TFs in the Terpenoid Pathway

Terpenoids synthesized from isoprene (C5) units also play diverse roles in plant–environment interactions [86]. AtMYC2-mediated positive regulation of the biosynthesis of these sesquiterpenes, the so-called phytoalexins, has been reported [87]. Methyl jasmonate (MeJA) and GA significantly increase the expression of the terpenoid pathway genes. AtMYC2 controls sesquiterpene biosynthesis in a JA-responsive manner through its interaction with GA signaling, especially through the competitive interaction of JAZ and DELLA in the JA and GA signaling cascades (Figure 1B).

### 3.3. TFs in the Camalexin Pathway

*A. thaliana* also produces an indole alkaloid, camalexin, as phytoalexin [88]. AtWRKY33 plays a crucial role in the biosynthesis of camalexin in response to pathogen attacks via the signaling cascade of mitogen-activated protein kinases (MAPKs) [89]. ANAC042, a member of the NAM, ATAF1/2, and CUC2 (NAC) TF family proteins, and MYB proteins have been reported to be involved in camalexin biosynthesis [72,90] (Figure 1C). The unique feature of camalexin biosynthesis is that the expression of camalexin biosynthetic genes is restricted to the tissue undergoing cell death [91]. The findings that cerato-platanin protein (an elicitor) produced by *Ceratocystis platani* triggers salicylic acid (SA)- and ET-signaling pathways, but not the JA signaling pathway, and induces the biosynthesis of camalexin in *A. thaliana*, indicated that camalexin biosynthesis is independent of JA signaling [92], whereas alkaloid biosynthesis in other pathways is largely dependent on JA signaling, as discussed below.

## 4. TFs Involved in The Regulation of Alkaloid Biosynthetic Enzyme Genes Expression

Alkaloid biosynthesis is regulated by several TF family proteins, such as bHLH and MYB, which are also found in the regulation of phenylpropanoid and terpenoid biosynthesis in *A. thaliana*; however, different types (groups) of TFs have been reported in several alkaloid-producing plant species. In this section, we describe TFs involved in the regulation of nicotine, SGA, MIA, and BIA biosynthesis.

### 4.1. TFs in the Nicotine and SGA Pathways

Nicotine is biosynthesized in the *Nicotiana* species by means of pyrrolidine ring formation and pyridine ring formation, followed by coupling of both rings [1]. Ornithine or arginine is converted to putrescine, which is then converted to *N*-methylputrescine, the first specific metabolite for the biosynthesis of nicotine, by putrescine *N*-methyltransferase (PMT). Aspartate is converted to nicotinic acid containing the pyridine ring by several enzymes, including quinolinic acid phosphoribosyltransferase (QPT). Although the mechanism of final condensation to produce nicotine is not enzymatically revealed, A622, a pinoresinol-lariciresinol reductase/isoflavone reductase/phenylcoumaran benzylic ether reductase family oxidoreductase, is required for the coupling.

To explore the TFs involved in nicotine biosynthesis, the cDNA-amplified fragment length polymorphism approach, in combination with target metabolite analysis, was performed using JA-treated tobacco Bright Yellow-2 (BY-2) cultured cells [93]. In these studies, two *AP2*/*ERF* genes, *NtORC1*/*ERF221* and *NtJAP1*/*ERF10*, were identified [12,94]. NtORC1/ERF221 and NtJAP1/ERF10 positively regulated the *PMT* gene, while the overexpression of *NtORC1*/*ERF221* only resulted in high nicotine alkaloid [13].

*AP2*/*ERF* genes were independently identified using microarray analysis of a *Nicotiana tabacum NIC2* mutant [11]. The *NIC2* locus includes seven clustered *AP2*/*ERF* genes belonging to the group IX AP2/ERF subfamily. Among them, the *NtERF189* gene, which encodes for subclade 2–1 of group IX AP2/ERF protein, showed clear responsiveness to MeJA and was expressed in roots, where nicotine is preferentially produced. RNA silencing of *NtERF189* significantly reduced the expression levels of nicotine biosynthetic enzyme genes, including *PMT* and *QPT*. The overexpression of *NtERF189* also showed a marked increase in the transcript levels of these genes, suggesting that NtERF189 functions as a master regulator of nicotine biosynthesis in tobacco (Figure 2A). Recently, the overexpression of *NtERF189* in stable transgenic tobacco plants has been analyzed [95]. These plants showed an increase in the total alkaloid content in leaves, up to 4.3- to 17.5-fold; in addition, the plant growth was also affected, possibly due to the drastic alteration in the contents of nitrogen-containing compounds. The NtERF189 protein recognizes GCC-box-like elements, (A/C)GC(A/C)(A/C)NCC, in the promoter of nicotine biosynthetic enzyme genes [96]. Interestingly, NtERF189-binding sites were found in the promoter of the MeJA-responsive *QPT2* gene involved in nicotine biosynthesis, but not in that of the *QPT1* gene which is not induced by MeJA. This suggests that ERF189-binding sites might have been acquired due to gene duplication in the *QPT2* promoter during evolution [97].

Another *AP2*/*ERF* gene, *NtERF32* (*EREBP2*), has also been identified in tobacco plants. This gene encodes the group IX AP2/ERF protein, which is not a part of the *NIC2* locus. NtERF32 might function as a transcriptional activator to regulate the expression of multiple genes involved in nicotine biosynthesis by binding to the GCC-box-like element [14].

Several groups have reported that MYC2-type bHLH TFs are involved in the regulation of nicotine biosynthesis. Todd et al. performed functional screening and identified two MeJA-responsive bHLH TF genes, *NbbHLH1* and *NbbHLH2*, in *Nicotiana benthamiana* [25]. RNA silencing of *NbbHLH1* and *NbbHLH2* significantly decreased the expression levels of nicotine biosynthetic enzyme genes, whereas their overexpression slightly increased *PMT* expression and nicotine content. NbbHLH1 can bind to the G-box in the *PMT* promoter, which contains the target GCC-box of NtORC1/ERF221 proximal to this G-box. NbbHLH1 and NtORC1/ERF221 cooperatively regulate the expression of *PMT* by interacting with both the elements. NbbHLH1 interacts with AtJAZ1 and NtJAZ1, suggesting that JA-mediated regulation of nicotine biosynthesis happens through the MYC2-JAZ complex [13].

Highly homologous genes of *NbbHLH1* and *NbbHLH2* have also been identified in *N. tabacum*. NtMYC2a and NtMYC2b positively regulate the transcript levels of nicotine biosynthetic enzymes and transporter genes as well as *NIC2*-locus *AP2*/*ERF* genes [26,27,28]. The direct interaction between NtMYC2 and NtJAZ1 has also been confirmed. NtMYC2 forms a regulatory complex with NtJAZ1 and controls the expression of both nicotine biosynthesis genes and other *AP2*/*ERF* genes in JA signaling.

SGAs are commonly produced by plants belonging to the Solanaceae family. The biosynthetic pathway starts with the precursor cholesterol and proceeds via hydroxylation, oxidation, glycosylation, and transamination steps to generate various SGAs [1]. The genomes of tomato and potato revealed that clustered genes named *glycoalkaloid metabolism* (*GAME*) are involved in the central pathway of SGAs.

Two research groups identified a novel AP2/ERF transcription factor, GAME9/JA-responsive ethylene response factor 4 (JRE4), which is involved in the regulation of SGA biosynthesis in tomato and potato [15,16]. *GAME9*/*JRE4* was co-expressed with 37 genes involved in SGA biosynthesis and was highly expressed in leaves, buds, and immature green fruit. *SlGAME9*-RNA interference (RNAi) and *jre4_1* mutant tomato plants showed decreased SGA content and transcript levels of biosynthetic enzyme genes [15,98]. Overexpression of *GAME9* altered the expression of genes involved in sterol and SGA biosynthesis (Figure 2B). Although *GAME9*/*JRE4* was clustered with four homologous genes, only GAME9/JRE4 was confirmed to control SGA biosynthesis. Furthermore, SlMYC2 and SlGAME9/JRE4 might bind to the GCC-box and G-box in the proximal region of promoters and coordinately regulate SGA biosynthesis genes in tomato. Such synergistic regulation of pathway genes by group IX AP2/ERF and MYC2 TFs is shared in nicotine and SGA biosynthesis.

### 4.2. TFs in the MIA Pathway

MIAs are biosynthesized by means of two primary metabolic routes: the shikimate and methylerythritol phosphate (MEP) pathways [1]. Tryptamine derived from tryptophan and secologanin derived from geranyl diphosphate (GPP) are condensed to strictosidine, the central intermediate of MIAs, by strictosidine synthase (STR). Strictosidine is deglucosylated by strictosidine glucosidase (SGD) and then converted to diverse MIAs, such as serpentine, catharanthine, vindoline, and anti-tumor bisindole alkaloids (vinblastine and vincristine).

The first alkaloid TFs to be identified in *C. roseus* using yeast one-hybrid screening were AP2/ERF TFs, which were named octadecanoid-derivative responsive *Catharanthus* AP2-domain (ORCA) 1 and ORCA2 [4]. ORCA1 binds to the JA-responsive element (JRE) of the *STR* promoter but has little transcriptional activity, whereas ORCA2 not only binds to the JRE but also shows in vivo trans-activation activity and JA-responsive expression. The overexpression of *ORCA2* in hairy roots also suggests a key function of ORCA2 in MIA biosynthesis [99]. ORCA3, another MeJA-responsive AP2/ERF TF, was identified in *C. roseus* using an activation-tagging approach [5,100]. ORCA3 induces the expression of several MIA biosynthetic genes, including *STR*. Hence, ORCA3 is considered a master regulator in MIA biosynthesis, but it is not sufficient for the complete regulation of the MIA biosynthetic pathway (Figure 3).

More recently, *ORCA4*, *ORCA5*, and *ORCA6* have been reported to be present in the same genomic region as *ORCA2* and *ORCA3* [6,7,8]. All clustered *ORCA* genes were upregulated in response to MeJA, but their expression patterns were slightly different. Using transient reporter assay, ORCA4, ORCA5, and ORCA6 were found to trans-activate the *STR* promoter, similar to ORCA2 and ORCA3. Overexpression of *ORCA4* and *ORCA5* in *C. roseus* hairy roots and transient overexpression of *ORCA6* in *C. roseus* flower petals resulted in the upregulation of MIA biosynthetic enzyme genes as well as other TF genes, including *ORCA* genes, and a significant increase in the accumulation level of several MIAs. Clustered ORCA TFs regulate the expression of different sets of MIA biosynthetic genes, suggesting that their function might overlap and diversify during evolution.

In addition to ORCA TFs, a few AP2/ERF TFs have also been identified in *C. roseus*. CrERF5 belongs to a different clade from ORCA TFs in the phylogenetic tree and activates the MIA biosynthetic pathway, especially the bisindole alkaloid pathway [9]. The expression of *CrERF5* was induced in response to MeJA and ET, suggesting that ET signaling might affect the production of bisindole alkaloids. *Catharanthus roseus* 1 (CR1) was also identified in *C. roseus* and characterized as a transcriptional repressor of MIA biosynthesis. The expression of *CR1* decreased in response to MeJA, and the suppression of *CR1* showed an increase in the accumulation of serpentine and vindoline [10]. These *AP2*/*ERF* genes were characterized using virus-induced gene silencing (VIGS), an effective tool for the investigation of gene function.

CrMYC1 was the first bHLH TF to be reported for MIA biosynthesis [29]. Although the binding activity of CrMYC1 to the G-box sequence in the *STR* promoter was confirmed, there was no further characterization to show that CrMYC1 plays a role in the regulation of MIA biosynthesis in *C. roseus*. Zhang et al. have identified another bHLH TF, CrMYC2, in *C. roseus* [30]. CrMYC2 is homologous to AtMYC2 and regulates the expression of *ORCA2* and *ORCA3*, leading to the regulation of MIA biosynthesis. It also regulates the expression of *CrJAZ* genes and directly interacts with CrJAZ proteins that repress the function of CrMYC2 [36]. Upon overexpression of *CrMYC2*, the expression levels of *ORCA4* and *ORCA5* were also upregulated; however, CrMYC2 could not directly trans-activate the *ORCA4* and *ORCA5* promoters, suggesting that unknown TFs might play a role in regulating the expression of these genes [6]. While CrMYC2 directly binds to the promoter of the *tryptophan decarboxylase* (*TDC*) gene, a key gene for tryptamine biosynthesis, the expression of *STR* is indirectly regulated via ORCA3.

Based on the classification of plant bHLH proteins, MYC2 belongs to group IIIe [57]. Recently, non-MYC2 type bHLH proteins, which belong to group IVa, have been identified in *C. roseus* and designated as bHLH iridoid synthesis (BIS) [31]. *BIS1*, *BIS2*, and *BIS3* genes are clustered in the same genomic region of *C. roseus*, and their expression can be induced using MeJA. BIS1 and BIS2 form homo- or heterodimers, while a clear interaction with CrMYC2 has not been observed. BIS proteins regulate the expression of genes involved in the secoiridoid pathway [31,32,33].

A MeJA-responsive *WRKY* gene, *CrWRKY1*, was identified in *C. roseus* using degenerate PCR [20]. CrWRKY1 belongs to the Group III subfamily, and overexpression of *CrWRKY1* caused an increase in the expression of *TDC* gene and decrease in those of *ORCA2* and *ORCA3* genes, thus suggesting that CrWRKY1 might preferentially regulate serpentine biosynthetic pathway by activating *TDC* as well as repressing *ORCA* genes.

*CrBPF-1* gene, which encodes a MYB-like box-P binding factor protein, was identified using yeast one-hybrid screening with the *STR* promoter region containing an elicitor-responsive element [37]. Since *CrBPF-1* gene was significantly induced by an elicitor, but not MeJA, CrBPF-1 might function through JA-independent elicitor signaling. Overexpression of *CrBPF-1* significantly upregulated the expression of MIA biosynthesis genes while modestly affecting the accumulation of metabolites [38].

Vom Endt et al. identified five DNA-binding AT-hook motif-containing proteins in *C. roseus*. AT-hook proteins could trans-activate the expression of the *ORCA3* promoter, whereas the expression of *AT*-*hook* genes was not altered upon MeJA treatment, indicating that AT-hook proteins may act as activators independent of JA signaling [43].

Light is an important environmental cue for converting tabersonine to vindoline in *C. roseus*. Recently, light-induced GATA-type TF, CrGATA1, was identified and confirmed to be related to the alteration of gene expression involved in vindoline production in response to light [42]. Furthermore, the expression of *CrGATA1* is negatively controlled by the light-responsive phytochrome-interacting factor (PIF) family TF, CrPIF1.

Transcriptional repressors have also been identified in *C. roseus* and are considered to be involved in the modulation of MIA biosynthesis (Figure 3). Two G-box binding factor (GBF) proteins, CrGBF1 and CrGBF2, both containing the bZIP motif, possibly act as transcriptional repressors of MIA biosynthesis [40]. When *GBF1* and *GBF2* genes were co-expressed with *CrMYC2*, these proteins directly interacted with CrMYC2, suggesting that CrGBF proteins possibly act as antagonists of CrMYC2 and fine-tune gene expression in MIA biosynthesis [101].

Pauw et al. identified three genes that encode TFIIIA-type zinc finger proteins and named them *ZCT1*, *ZCT2*, and *ZCT3*, for zinc finger *Catharanthus* transcription factor [41]. RNA silencing of *ZCT1* had little effect on the expression of MIA biosynthetic enzyme genes and production of MIA, indicating that the three ZCT repressors might function redundantly [102]. ZCT proteins act as transcriptional repressors of *STR* and *TDC* promoters. Since CrMYC2 and ORCA proteins activate the expression of *ZCT* genes [6,7,8,101], ZCT proteins might fine-tune the spatiotemporal expression of genes involved in MIA biosynthesis.

In *A. thaliana*, MYC2-like group IIId bHLH TFs are known to repress the expression of MYC2-target genes. The Repressor of MYC2 Target 1 (RMT1) was identified in *C. roseus* and confirmed to repress the expression of *ORCA3* [36]. The expression of *RMT1* was induced by MeJA and CrMYC2. RMT1 did not interact with CrMYC2 but directly bound to the AACGTG sequence in the *ORCA3* promoter, which is a target of MYC2, suggesting that RMT1 represses the *ORCA3* gene by competing with CrMYC2 to bind to the same *cis*-element.

Although several TFs have been characterized in the MIA pathway, the characterization of TFs involved in cell differentiation is limited. Since vindoline biosynthesis occurs in laticifer/idioblast cells, understanding the TFs involved in laticifer/idioblast differentiation is crucial for the production of vindoline in undifferentiated cells. Several TFs involved in laticifer differentiation have been characterized in *Hevea brasiliensis*; these include MYB TFs (identified using transcriptome analysis) and an AP2/ERF TF, HbEREBP1 as a negative regulator of defense genes in laticifers [103,104].

### 4.3. TFs in the BIA Pathway

BIA biosynthesis starts with the conversion of L-tyrosine to dopamine and 4-hydroxyphenylacetaldehyde, followed by condensation to (*S*)-norcoclaurine, the central precursor of BIAs. (*S*)-norcoclaurine is converted to (*S*)-reticuline, the central intermediate of BIAs, by three methyltransferases and one cytochrome P450, and (*S*)-reticuline is converted to various BIAs such as analgesic morphine, antimicrobial berberine, and sanguinarine [1,105]. Almost all of the enzyme genes involved in the biosynthesis of these alkaloids have been identified and characterized, in addition to which metabolic engineering approaches have been attempted to manipulate the BIA biosynthetic pathways [1]. Furthermore, whole-genome sequencing of several BIA-producing plants such as *P. somniferum*, *E. californica*, *Aquilegia coerulea*, *Nelumbo nucifera*, *Macleaya cordata*, and *Coptis chinensis* has been performed, which provides us with more information for the identification of unknown genes involved in BIA biosynthesis [106,107,108,109,110,111,112].

A group IIc WRKY TF, CjWRKY1, was first identified in *C. japonica* using transient RNAi screening as a transcriptional activator in berberine biosynthesis [19] (Figure 4). The *CjWRKY1* gene showed clear MeJA-responsiveness and *CjWRKY1* regulated almost all BIA biosynthetic enzyme genes by binding to the W-box in the promoter of target genes in *C. japonica* cells. Heterologous expression of *CjWRKY1* in *E. californica* (California poppy), which shares a common BIA biosynthetic pathway, resulted in increased BIA production, although the expression levels of several BIA biosynthetic enzyme genes were not altered [113]. A recent genome-wide analysis of California poppy *WRKY* genes suggested that the WRKY TF family might be functionally diversified in BIA biosynthesis and associated with the accumulation and translocation of BIAs [114].

In *P. somniferum*, wound-induced PsWRKY, which belongs to the group I WRKY protein, was found to be involved in the BIA biosynthetic pathway [21]. PsWRKY can bind to the W-box DNA sequence and trans-activate the *tyrosine decarboxylase* (*TYDC*) gene promoter. Apuya et al. revealed that heterologous expression of *A. thaliana WRKY1* (*AtWRKY1*), which also belongs to the group I WRKY protein, enhanced the production of BIAs in *P. somniferum* and *E. californica* cells [115]. These results indicate that group I WRKY TFs might also be involved in the regulation of BIA biosynthesis.

Transient RNAi screening in *C. japonica* cells further highlighted the bHLH TF, CjbHLH1, as a comprehensive transcriptional activator in BIA biosynthesis [34]. While MYC2-type bHLH TFs regulate the biosynthesis of nicotine and MIA, CjbHLH1 is a non-MYC2-type bHLH TF, as deduced based on the clear difference in amino acid sequence and length [69]. Interestingly, homologous proteins of CjbHLH1 are only found in BIA-producing plant species, and EcbHLH1-1/1-2, which are CjbHLH1 homologs of California poppy, have been confirmed to regulate BIA biosynthesis in California poppy cells [35]. These results suggest that a unique type of bHLH TF plays a key role in the regulation of BIA biosynthesis in BIA-producing plants, and functionally diverse bHLH TFs form different transcriptional networks in the biosynthesis of different types of alkaloids. Although the expression of *CjbHLH1* and *EcbHLH1-1*/*1-2* is induced by MeJA, the functional relationship between non-MYC2-type bHLH and MYC2 remains unclear.

Characterization of *cis*-acting elements in the promoter region of BIA biosynthetic enzyme genes in *C. japonica* cells indicated the involvement of AP2/ERF TF protein(s) in the regulation of BIA biosynthesis [116]. Genome-wide investigation of AP2/ERF family genes in the *E. californica* genome revealed that several Group IX *EcAP2*/*ERF* genes show a clear response to MeJA. Furthermore, four Group IX AP2/ERF TFs trans-activate BIA biosynthetic enzyme genes [18].

As discussed in the MIA pathway, morphine biosynthesis also occurs among several different cell types, including sieve elements and laticifer cells. The spatial accumulation of specific types of BIAs in opium poppy plants and cultured cells may be regulated by different networks of TFs. Furthermore, different organs produce different types of BIAs; the aerial part produces pavine-type BIA, while roots produce benzophenanthridine-type BIA in California poppy [114]. These organ-specific regulatory mechanisms of BIA biosynthesis have not been sufficiently characterized.

## 5. Regulatory Mechanism: Upstream Signals, i.e., JA-Mediated, and Post-Transcriptional Regulation

JAs are key signaling molecules involved in secondary metabolism, including alkaloid biosynthesis. As mentioned above, the JA signaling pathway has been intensively investigated in *A. thaliana*. MYC2, COI1, and JAZ form a regulatory complex with novel interactor of JAZ (NINJA), TOPLESS (TPL), and MEDIATOR 25 (MED25), and orchestrate the JA-signaling cascade; some homologs in alkaloid pathways have been isolated based on sequence homology [62].

Tobacco COI1 homologs, NaCOI1 and NtCOI1, have been identified in *Nicotiana attenuata* and *Nicotiana tabacum*, respectively [117,118]. RNA silencing of these genes revealed that COI1 plays an important role in nicotine production. Three *JAZ* genes have also been identified in *N. tabacum*, and the direct interaction of NtJAZ1 with NtMYC2 and NbbHLH1/NbbHLH2 has been confirmed [26,28,118]. Hence, it can be concluded that nicotine biosynthesis is regulated by the COI1-JAZ-MYC2 regulatory complex in the JA signaling cascade. Recently, a COI1 homolog has also been reported to be important for the regulation of SGA biosynthesis via GAME9/JRE4 in tomato [119].

The *C. roseus* COI1 homolog, CrCOI1, has also been reported to interact with several CrJAZ proteins [36]. RNA silencing of the *CrCOI1* gene resulted in significant repression of the MeJA-induced upregulation of *ORCA3*, *CrMYC2*, *BIS1*, *BIS2*, and *geraniol 10*-*hydroxylase* (*G10H*) genes. In contrast, overexpression of *CrCOI1* increased the expression levels of biosynthetic enzymes and TF genes involved in MIA biosynthesis and the accumulation of MIAs. These results suggest that the COI1-JAZ-MYC2 complex might control the expression of several TF genes to modulate the metabolic pathway of MIA. On the other hand, the involvement of the COI1-JAZ-MYC2 core complex in BIA biosynthesis has not been confirmed, while its presence is suggested in JA signaling.

Protein phosphorylation has also been investigated as a mechanism for the regulation of alkaloid biosynthesis. In tobacco, the induction of *JA*-*factor*-*stimulating MAPKK1* (*JAM1*) by MeJA coincided with the expression of *NtORC1*/*ERF221*. Co-overexpression of *JAM1* with *NtORC1*/*ERF221* or *NbbHLH1* synergistically increased the expression of *PMT* and *QPT* promoters in response to MeJA [13,93]. These findings suggest that the MAPK phosphorylation cascade is involved in the regulation of nicotine biosynthesis through NtORC1/ERF221 and NbbHLH1.

The expression of MIA biosynthesis genes in response to MeJA was hindered by the addition of protein kinase inhibitors, suggesting the involvement of phosphorylation in the regulation of MIA biosynthesis. The MeJA-responsive *MPK3* gene has been identified in *C. roseus* [120]. Overexpression of *CrMPK3* increased the accumulation of several MIAs. More recently, MAP kinase kinase kinase (MAPKKK1), MAP kinase kinase (MAPKK), and MPK6 have also been identified in *C. roseus* [6]. The expression of *CrMAPKKK1* and *CrMAPKK1* was induced by MeJA, and CrMAPKKK1 interacted with CrMPK3 and CrMPK6. The trans-activation activity of ORCA2–6 and CrMYC2 was significantly enhanced by the co-expression of *CrMAPKK1* [8]. Overexpression of *CrMAPKK1* showed significant increases in the transcript levels of MIA biosynthetic enzyme genes and *ZCT1-3* as well as accumulation of tabersonine and catharanthine. These results suggest that the CrMPK cascade might be involved in the regulation of MIA biosynthesis via phosphorylation of CrMYC2 and ORCA TFs (Figure 3).

Tyrosine phosphorylation and protein degradation have been reported to be involved in the regulation of BIA biosynthesis in *C. japonica* [121]. The tyrosine phosphorylation of the WRKYGQK core domain in CjWRKY1 affected its DNA-binding activity and subcellular localization, subsequently resulting in CjWRKY1 not being able to activate the transcription of BIA biosynthesis genes. Phosphorylated CjWRKY1, which is localized in the cytosol, might be rapidly degraded by proteases. Furthermore, CjWRKY1, which is not phosphorylated, may also be degraded in the nucleus by the 26S proteasome. These findings indicate that BIA biosynthesis might be intensively controlled at the post-translational level, with a combination of protein phosphorylation and degradation (Figure 4).

## 6. TFs and Alkaloid Engineering

TFs are powerful tools for improving production yield and quality by controlling the overall expression of metabolic pathway genes. Since the application of TFs in alkaloid biosynthesis is limited due to a lack of information about the regulatory mechanisms involved, many researchers have tried to explore a master regulator of the biosynthetic pathway and elucidate its function and transcriptional network to achieve alkaloid engineering using several TFs.

For example, ectopic expression of *CjWRKY1* in cultured California poppy cells increased the accumulation of several BIAs such as sanguinarine, chelirubine, and chelerythrine in the culture medium, in addition to enhancing the total BIA content up to 2- to 5-fold [113]. The modest effect on alkaloid productivity is due to the post-translational modifications of CjWRKY1. The overexpression of TFs, such as *ORCA3* and *CrBPF-1* in *C. roseus* hairy roots, increased the expression levels of several MIA biosynthetic genes, whereas the accumulation levels of MIAs were only slightly altered even after MeJA treatment [38,122]. The expression of *ZCT* genes was also upregulated in these hairy roots, suggesting that the metabolic pathway involved might be rigorously fine-tuned by multiple activators and repressors. Thus, the overexpression of single and native TF genes might be limited to a modest effect on the flux of the biosynthetic pathway. A combinatorial induction of activator genes such as *ORCA* and *CrMYC2*, with the suppression of repressor genes such as *ZCT1-3*, would be useful in enhancing the metabolic flow of biosynthetic pathways more efficiently.

Interestingly, Apuya et al. demonstrated the heterologous expression of *AtWRKY1* in California poppy and opium poppy cells. Transgenic California poppy cells showed a marked increase in the accumulation of dihydrosanguinarine and 10-hydroxydihydrosanguinarine by up to 30-fold and 34-fold, respectively. Furthermore, the overexpression of *AtWRKY1* in opium poppy increased the thebaine content by up to five-fold [115]. These findings suggest that the regulatory TF network in alkaloid biosynthesis might be similar among plant species, but the possible post-translational modifications might be different. Exchange of ERF family TFs between nicotine and MIA pathways has also been reported [7].

Metabolic engineering targeting multiple biosynthetic enzymes and TF genes has also been reported. Wang et al. performed overexpression of *ORCA3* in combination with that of *G10H*, which is not controlled by ORCA3, in *C. roseus* hairy roots [123]. Transgenic roots showed increased accumulation of catharanthine and expression levels of *STR* and *secologanin synthase* (*SLS*). Co-overexpression of *ORCA3* and *SGD* in *C. roseus* hairy roots also increased the accumulation levels of several MIAs, such as serpentine and tabersonine, as well as the transcript levels of MIA biosynthetic enzymes and TF genes [124].

The transient overexpression of multiple TF genes, as examined by Schweizer et al. in *C. roseus* flower petals using agroinfiltration, would be a promising approach to evaluate the modified TF functions in the regulation of biosynthesis. The mutation of CrMYC2 at D126 affects its interaction with JAZ proteins, and CrMYC2^D126N^ strongly induces the expression of genes involved in MIA biosynthesis and transport. Furthermore, combinatorial overexpression of *CrMYC2^D126N^*, *BIS1*, and *ORCA3* synergistically increased the expression of iridoid and MIA pathway genes and the accumulation levels of several MIAs, including possibly root-specific hörhammericine and 16-methoxyhörhammericine [125].

The above engineering has been mainly performed in de-differentiated cultured cells. However, some alkaloids such as vindoline and morphine are produced by the collaboration of several different cells, such as leaf or root tissues, including epidermal cells, parenchyma cells, and laticifer/idioblast cells. Modification of cell differentiation and in vitro organ culture, especially that of hairy root, might be a more attractive approach to promote metabolic expression since in vitro culture does not require entire plant development. The control of environmental conditions, including light, temperature, and substrate supply, including artificially modified chemicals, could be optimized in vitro with genetic modification. The relationship between tissue/cell-specific expression of biosynthetic enzyme genes and JA-responsive TFs involved in alkaloid biosynthesis is not well studied; however, the detailed tissue/cell-specific regulation of alkaloid biosynthesis may contribute to a novel engineering approach for targeting cell differentiation. In *Artemisia annua*, an AP2/ERF TF, trichome and artemisinin regulator 1 (TAR1) was reported to regulate the development of trichomes and the biosynthesis of artemisinin, which is produced and stored in trichomes [126]. Recent advances in genetic tools might enable us to manipulate the proliferation of cells involved in the production and accumulation of specific metabolites in the future.

## 7. Conclusions

TFs are central regulators that regulate the expression of genes involved in plant developmental processes and biotic and/or abiotic stress responses. Progress in molecular biological techniques and recent advances in next-generation sequencing technologies have enabled us to perform a comprehensive analysis of TF genes and to explore regulators of alkaloid biosynthesis in several plant species. Indeed, different types of TFs, such as AP2/ERF, WRKY, and bHLH, have been identified in the biosynthesis of each alkaloid and regulatory network, including the MAPK cascade; in addition, post-translational modifications of TFs have also been identified. Some of them are widely distributed in the model plants and are involved in the regulation of secondary metabolism or other plant processes, for example, the MYC2-type bHLH TFs function with COI1 and JAZ proteins in the JA signaling cascade and modulate the expression of genes involved in alkaloid, terpenoid, and anthocyanin biosynthesis. Others are specifically found in alkaloid-producing plants, such as non-MYC2-type bHLH TFs in BIA biosynthesis. A comparison of TFs involved in alkaloid biosynthesis with those in terpenoid and phenylpropanoid indicates that the regulatory mechanisms shared in many plants and unique TFs might be evolved in a coordinated manner to form a regulatory network for each biosynthetic pathway. The genome information of many plant species provides some insights into the evolution of transcriptional regulatory networks. *AP2*/*ERF* gene clusters have been found in the genomes of *C. roseus*, *N. tabacum*, *S. lycopersicum*, *S. tuberosum*, and *E. californica*, but their functions might have differentiated during evolution [18,48]. The evolutionary and biological significance of clustered TF genes is not fully understood; however, biosynthetic genes encoding several classes of enzyme proteins involved in plant-specialized metabolism are often clustered, and their regulation might be involved in the duplicated TFs. An understanding of the regulatory mechanisms associated with clustered TFs in alkaloid biosynthesis, in comparison to those of phenylpropanoid and terpenoid biosynthesis, should be helpful for elucidating the evolution of plant-specialized metabolism and manipulating the specific metabolic pathways resulting in the efficient production of specific metabolites. Furthermore, combinatorial genetic manipulation, such as genome editing and protein modification based on the structure and post-translational regulatory mechanisms, may contribute to a full understanding of the complicated regulatory mechanisms and the development of systems for the efficient production of many valuable metabolites in plants. An elucidation of tissue/cell-specific regulation of genes involved in alkaloid biosynthesis also has the potential to manipulate specific metabolic pathways. Further characterization of TFs that might regulate the expression of genes involved in the compartmentation of valuable metabolites and cell differentiation will be required.

## Figures and Tables

**Figure 1 biomolecules-11-01719-f001:**
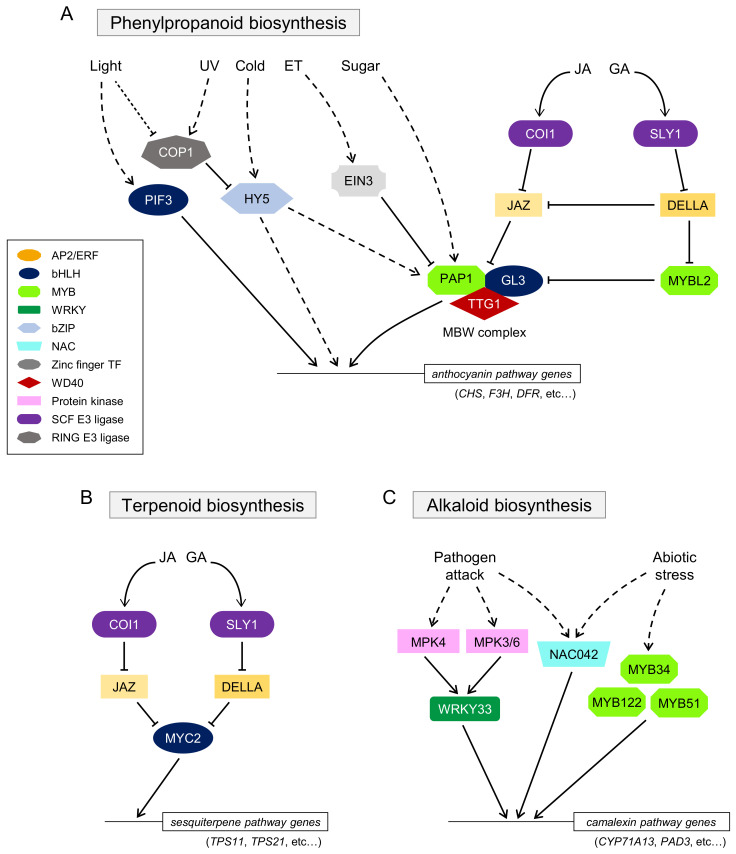
A brief model of the regulation of phenylpropanoid (**A**), terpenoid (**B**), and camalexin (**C**) biosynthesis in *A. thaliana*. Black arrows indicate upregulation, while T-shaped lines indicate inhibition. Broken lines show indirect or undetermined steps that are possibly involved in uncharacterized proteins or unidentified regulation. JAZ and DELLA include the functionally redundant proteins. Abbreviations: CHS, Chalcone synthase; COI1, Coronatine-insensitive 1; COP1, Constitutive photomorphogenesis 1; DFR, Dihydro flavonol reductase; EIN3, Ethylene-insensitive 3; ET, Ethylene; F3H, Flavanone 3-hydroxylase; GA, Gibberellin; GL3, Glabra 3; HY5, Elongated Hypocotyl 5; JA, Jasmonate; JAZ, Jasmonate ZIM domain; MBW, MYB–bHLH–WD40; MYBL2, MYB-LIKE2; PAD3, Phytoalexin-deficient 3; PAP1, Production of Anthocyanin Pigment 1; PIF3, Phytochrome-interacting factor 3; TPS, terpene synthase; TTG1, Transparent Testa Glabra 1; SLY1, Sleepy 1.

**Figure 2 biomolecules-11-01719-f002:**
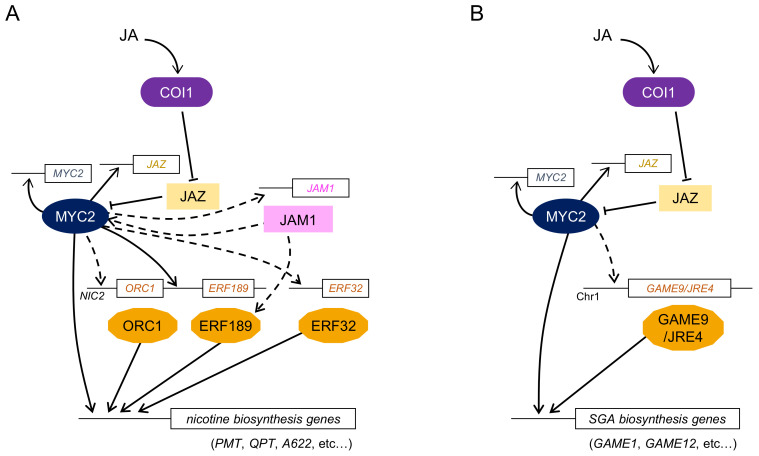
A simplified model of the transcriptional network of JA signaling in nicotine biosynthesis in tobacco (**A**) and SGA biosynthesis in tomato (**B**). Boxes with straight lines indicate regulator genes, including promoter regions. Black arrows indicate upregulation, while T-shaped lines indicate inhibition. Broken lines show indirect or undetermined steps that are possibly involved in uncharacterized proteins or unidentified regulation. Possible feedback regulation via MYC2 TF, which is known to regulate the expression of several *JAZ* genes, is shown. In tobacco, functionally redundant MYC2 proteins such as NtMYC2a, NtMYC2b, NbbHLH1, and NbbHLH2 have been identified; of these, they are collectively indicated with MYC2. *ORC1* and *ERF189* genes and *GAME9*/*JRE4* are clustered in the *NIC2* locus in *N. tabacum* and chromosome 1 in tomato, respectively, with other group IX *AP2*/*ERF* genes. Abbreviations: GAME, Glycoalkaloid metabolism; JAM1, JA-factor-stimulating MAPKK1; JRE4, JA-responsive ethylene response factor 4; PMT, Putrescine *N*-methyltransferase; QPT, Quinolinic acid phosphoribosyltransferase; SGA, Steroidal glycoalkaloid.

**Figure 3 biomolecules-11-01719-f003:**
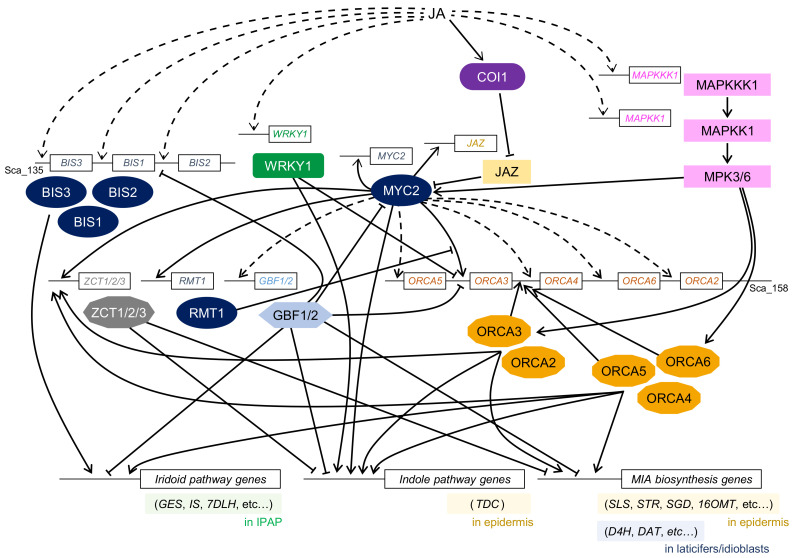
A simplified model of the transcriptional network of JA signaling involved in MIA biosynthesis in *C. roseus*. Boxes with straight lines indicate regulator genes, including promoter regions. Black arrows indicate upregulation, and T-shaped lines indicate inhibition. Broken lines show indirect or undetermined steps possibly involved in uncharacterized proteins or unidentified regulation. GES, IS, and 7DLH marked in the green box work in the iridoid pathway to synthesize loganic acid in internal phloem-associated parenchyma (IPAP), while D4H and DAT marked in the blue box function in the biosynthesis of vindoline in laticifers/idioblasts. The relationship between tissue/cell-specific regulation of enzyme genes and JA-responsive TFs remains unknown. Secologanin, tryptamine, strictosidine, tabersonine, and desacetoxyvindoline are synthesized in the upper or lower epidermis by SLS, STR, SGD, 16OMT, and TDC marked in the yellow boxes. Possible feedback regulation via MYC2 TF, which is known to regulate the expression of its own and *JAZ* genes, is shown. It is still unknown whether the expression of other genes such as *BIS*, *WRKY1*, and *MAPKKK1* is directly regulated by MYC2. *BIS* and *ORCA* genes are clustered in the same *C. roseus* genome scaffolds, scaffold_135 and scaffold_158, respectively. Abbreviations: BIS, bHLH iridoid synthesis; DAT, Deacetylvindoline 4-*O*-acetyltransferase; D4H, Desacetoxyvindoline 4-hydroxylase; GBF, G-box binding factor; GES, Geraniol synthase; IS, Iridoid synthase; MIA, Monoterpenoid indole alkaloid; ORCA, Octadecanoid-derivative responsive *Catharanthus* AP2-domain; RMT, Repressor of MYC2 Target 1; SGD, Strictosidine glucosidase; SLS, Secologanin synthase; STR, Strictosidine synthase; TDC, Tryptophan decarboxylase; 16OMT, 16-hydroxytabersonine *O*-methyltransferase; 7DLH, 7-deoxyloganic acid hydroxylase.

**Figure 4 biomolecules-11-01719-f004:**
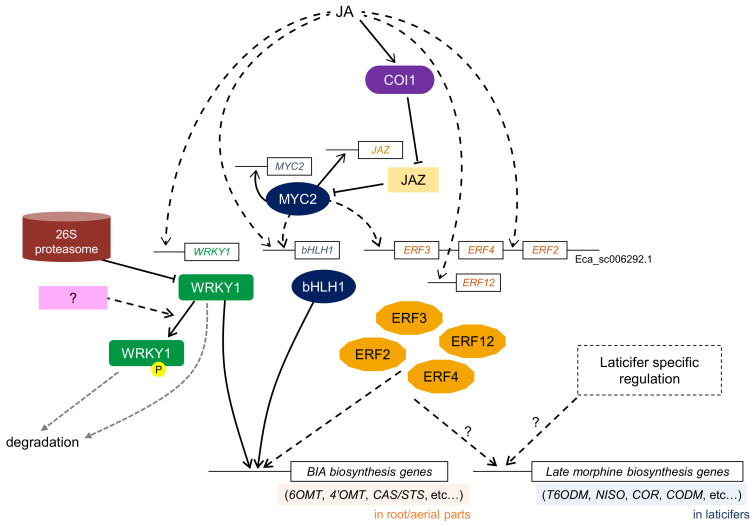
A simplified model of the transcriptional network of JA-signaling in BIA biosynthesis in *C. japonica* cultured cells and *E. californica* roots and cultured cells. Boxes with straight lines indicate regulator genes, including promoter regions. Black arrows and T-shaped lines indicate upregulation and inhibition, respectively. Broken lines show indirect or undetermined steps that are possibly involved in uncharacterized proteins or unidentified regulation. Gray dashed lines indicate degradation steps. Biosynthetic enzyme genes involved in late morphine pathway such as *T6ODM*, *NISO*, *COR*, and *CODM* might be controlled by laticifer-specific regulation, which remains to be determined. Since the functions of COI1, JAZ, and MYC2 proteins in BIA biosynthesis are not fully characterized, it is still unknown whether the expression of *WRKY1*, *bHLH1*, and several *ERF* genes is regulated by MYC2. Unphosphorylated WRKY1 is degraded by 26S proteasome, while unknown tyrosine kinase(s) and protease(s) might be involved in the phosphorylation and degradation of WRKY1, respectively, in *C. japonica*. BIA biosynthesis is regulated by several non-MYC2-type bHLH TFs (CjbHLH1 in *C. japonica* and EcbHLH1-1/2 in *E. californica*), while only bHLH1 is described here. *ERF2*, *ERF3*, and *ERF4* genes are clustered in the same *E. californica* genome scaffold, Eca_sc006292.1. Abbreviations: BIA, Benzylisoquinoline alkaloid; CAS/STS, (*S*)-canadine synthase/(*S*)-stylopine synthase; CODM, Codeine *O*-demethylase; COR, Codeinone reductase; NISO, Neopinone isomerase; T6ODM, Thebaine 6-*O*-demethylase; 6OMT, (*S*)-norcoclaurine 6-*O*-methyltransferase; 4’OMT, (*S*)-3’-hydroxy-*N*-methylcoclaurine 4’-*O*-methyltransferase.

**Table 1 biomolecules-11-01719-t001:** TFs that have been identified in alkaloid biosynthesis.

Family	Name	Accession No. or Gene ID	Plant Species	Reference
AP2/ERF	ORCA2	AJ238740	*Catharanthus roseus*	[4]
	ORCA3	EU072424	*Catharanthus roseus*	[5]
	ORCA4	KR703577	*Catharanthus roseus*	[6]
	ORCA5	KR703578	*Catharanthus roseus*	[7]
	ORCA6	MN614455	*Catharanthus roseus*	[8]
	CrERF5	MK862158	*Catharanthus roseus*	[9]
	CR1	cra_locus_10963	*Catharanthus roseus*	[10]
	NtERF189	AB827951	*Nicotiana tabacum*	[11]
	NtERF221/ORC1	CQ808982	*Nicotiana tabacum*	[12,13]
	NtERF32	AB828154	*Nicotiana tabacum*	[14]
	GAME9/JRE4	Solyc01g090340	*Solanum lycopersicum*	[15,16]
	OpERF2	LC171328	*Ophiorrhiza pumila*	[17]
	EcERF2	Eca_sc006292.1_g0200.1	*Eschscholzia californica*	[18]
	EcERF3	Eca_sc006292.1_g0150.1	*Eschscholzia californica*	[18]
	EcERF4	Eca_sc006292.1_g0190.1	*Eschscholzia californica*	[18]
	EcERF12	Eca_sc194641.1_g1370.1	*Eschscholzia californica*	[18]
WRKY	CjWRKY1	AB267401	*Coptis japonica*	[19]
	CrWRKY1	HQ646368	*Catharanthus roseus*	[20]
	PsWRKY	JQ775582	*Papaver somniferum*	[21]
	OpWRKY1	Opuchr09_g0007470-1	*Ophiorrhiza pumila*	[22]
	OpWRKY2	Opuchr02_g0001210-1	*Ophiorrhiza pumila*	[23]
	OpWRKY3	Opuchr09_g0002440-1	*Ophiorrhiza pumila*	[24]
bHLH	NbbHLH1	GQ859152	*Nicotiana benthamiana*	[25]
	NbbHLH2	GQ859153	*Nicotiana benthamiana*	[25]
	NtMYC2a	HM466974	*Nicotiana tabacum*	[26,27]
	NtMYC2b	HM466975	*Nicotiana tabacum*	[26,28]
	CrMYC1	AF283506	*Catharanthus roseus*	[29]
	CrMYC2	AF283507	*Catharanthus roseus*	[30]
	SlMYC2	NM_001324483	*Solanum lycopersicum*	[15]
	CrBIS1	KM409646	*Catharanthus roseus*	[31]
	CrBIS2	KM409645	*Catharanthus roseus*	[32]
	CrBIS3	MN646782	*Catharanthus roseus*	[33]
	CjbHLH1	AB564544	*Coptis japonica*	[34]
	EcbHLH1-1	AB910896	*Eschscholzia californica*	[35]
	EcbHLH1-2	AB910897	*Eschscholzia californica*	[35]
	RMT1	KY851107	*Catharanthus roseus*	[36]
MYB	BPF-1	AJ251686	*Catharanthus roseus*	[37,38]
	OpMYB1	LC076107	*Ophiorrhiza pumila*	[39]
bZIP	GBF1	AF084971	*Catharanthus roseus*	[40]
	GBF2	AF084972	*Catharanthus roseus*	[40]
TFIIIA zinc finger	ZCT1	AJ632082	*Catharanthus roseus*	[41]
	ZCT2	AJ632083	*Catharanthus roseus*	[41]
	ZCT3	AJ632084	*Catharanthus roseus*	[41]
GATA	CrGATA1	CRO_T134526	*Catharanthus roseus*	[42]
AT-hook	2D328	EF025306	*Catharanthus roseus*	[43]
	2D173	EF025307	*Catharanthus roseus*	[43]
	2D449	EF025308	*Catharanthus roseus*	[43]
	2D38M	EF025309	*Catharanthus roseus*	[43]
	2D7	EF025310	*Catharanthus roseus*	[43]

## Data Availability

No datasets were generated during the current work.

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
