# Peer review of "Transcription Factors in Alkaloid Engineering"

_biomolecules, 2021, doi:10.3390/biom11111719_

Round 1

Reviewer 1 Report

Manuscript Biomolecules-1441670 reviews research on transcription factors (TFs) involved in alkaloid biosynthesis is various plant species. The manuscript gives an insightful overview of transcription factors that have been found and provides clear models that summarize the transcription factor networks and how they function in signaling especially in response to the hormone jasmonic acid. Apart from small spelling mistakes English language is good. Below is a list of comments.

Major comments

  1. Lines 17-18, 600-601: it is suggested that new techniques and whole genome sequencing has led to the majority of current knowledge about transcription factors in alkaloid biosynthesis. In my view that is incorrect, for example Table 1 shows that in C. roseus 15 out of a total of 25 TFs were characterized before whole genome or transcriptome sequences were available. In addition, recent work in C. roseus with whole genome sequence databases has provided very little new information. The new genes are mainly homologues of the previously known genes, and the new knowledge about functions is mainly confirmatory and interesting for completionists only, as can also be seen from the fact that all these more recent papers were published in relatively low impact journals.
  2. Line 41: I am not sure there is a delay in knowledge about alkaloid biosynthesis as stated here and at several other places. It is true that knowledge about anthocyanin biosynthesis was boosted enormously by genetics in Arabidopsis, petunia, snapdragon and maize. But biochemically speaking alkaloid biosynthesis has been thoroughly studied for decades and the enzymatic knowledge on alkaloids is probably the best there is. And the ORCAs were the first TFs involved in JA responses discovered in any system, so certainly not a delay there. All enzyme genes in the MIA pathway are now known, and there are not so many secondary metabolite pathways for which this can be said.
  3. Paragraph 1.2: Description of the WRKY domain is very confusing. First in line 78-79 the WRKY domain is defined as consisting of a WRKY sequence and a zinc finger, which is correct. But then it says that the 3 groups differ in the number of WRKY domains and zinc fingers. This should the phrased differently, for instance 2 WRKY sequences and 2 zinc fingers etc (which I don’t like too much since that is just 2 WRKY domains), or 2 WRKY domains with a C2H2 type zinc finger, 1 WRKY domain with a C2H2 finger or 1 WRKY domain with a C2HC finger.
  4. Line 125: “in addition” should be removed, it should be “a MYB domain occurring in one to four copies”.
  5. Line 221, 299, 553, 577: Steroid induced or alcohol induced is confusing since it is not explained what the expression system is (could be endogenous steroids for example) and the information is actually irrelevant. Just use the term “overexpression” or “transgenic overexpression”.
  6. Line 231 and 232: I never understand these evolutionary explanations and usually find them completely irrelevant. Why did the QPT1 gene not lose its ERF binding site and thereby ceased to be involved in nicotine biosynthesis?
  7. Fig 2: Solid line from JA to MYC2 and JAZ promoters is confusing and wrong. It makes the model of JA regulation confusing suggesting that there are multiple receptors etc. From JA there should only a solid line to COI1. MYC2 regulates its own expression and JAZ expression, so there should be solid lines from MYC2 protein to MYC2 and JAZ promoters. Similar thing holds for Fig.3, but in that case it is not known whether all these promoters are regulated by MYC2. So either a dotted line from the MYC2 protein or a dotted line from JA to the promoters with a short explanation in the legend. Similar thing holds for Fig.4, COI1 and JAZ are even missing here. So dotted lines from JA to promoters? With some text in the legend saying that JA affects promoter activity probably via COI1, JAZ members and unidentified TFs?
  8. Line 287, 288: The secoiridoid pathway is not primary but secondary. The MEP pathway on the other hand is primary. The shikimate pathway is defined as resulting in chorismate. The pathway from chorismate to tryptophan is sometimes called the tryptophan pathway.
  9. 3: In my view the message conveyed about tissue specificity and JA regulation in this figure is totally wrong. The tissue-specific expression of genes was studied in plants that were not treated with JA. In my opinion JA would switch on all genes in a single cell type irrespective of tissue-specific regulation, based on the fact that genes in the secoiridoid pathway (G10H), middle part of the pathway (TDC, STR) and the vindoline branch (D4H, DAT) were all switched on in undifferentiated suspension cells treated with MeJA (van der Fits and Memelink, 2000). Authors might reconsider text in lines 405-411. Also they do not mention TFs involved in epidermis and IPAP differentiation. In the differentiation model, TFs involved in all these differentiations would be important and none are known currently. But as I mentioned before, in the JA model differentiation does not matter.
  10. Lines 580-597: In line with previous remark, authors may want to take a good look at this text although in principle it is not wrong and I agree (partly) with it.

Minor comments

  1. Line 15: What do the authors mean by “limited number of plant species produce alkaloids”? It is estimated that 10-15% of vascular plant species make alkaloids, that is 40,000-60,000 species. In all known cases regulation of alkaloid biosynthesis is very complex. In my opinion, the main reason that there is a delay (if there is any) is that only very few research groups were interested in transcription factors involved in alkaloid metabolism 1-2 decades ago.
  2. Line 65: the DNA elements and the TFs have been switched around.
  3. Line 155: a bit confusing since PAP1 is part of the MBW complex.
  4. Line 193: DFR is short for dihydro flavonol reductase.
  5. Line 209: replace “genes” by “transcription factors”.
  6. Line 263: replace “only MYC2 is described” by “they are collectively indicated with MYC2” if that is what the authors mean.
  7. Line 300: “subsequently” is out of place at this position since ORCA3 was long described before the function of ORCA2 in hairy roots was described.
  8. Line 342, 343: BIS TFs also regulate all genes in the MEP pathway.
  9. 3 bottom middle: What does “TDC etc” mean? There is only one gene involved in the formation of the indole moiety and that is TDC.
  10. Line 372: Vom Endt described 5 AT-hook proteins.
  11. Line 455: TYDC has not been defined.
  12. Line 496: mistake in N. attenuata. Common due to automatic spelling correction
  13. Line 554: mistake in C. roseus.
  14. Line 612: Mistake in E. californica.

Author Response

Thank you for your constructive comments.

Reviewer 2 Report

Dear authors, your manuscript looks good for publicación, although I suggest You some minor revision before acceptance.

  1. I suggest that sección 2 (regarding TFs in phenylpropanoids and terpenes) must be used as a final comparison, once You described (as You do very well) the aspects with alkaloids.
  2. Ir woyld be interesting mentioning some examples using both gene silencing and transient expression using VIGS. As a section.

Author Response

(The authors gave the same response as above.)

Reviewer 3 Report

Review responses:

This review article discusses transcription factors involved in secondary metabolite biosynthesis. Predominantly, authors explained TFs involved in regulating alkaloid biosynthetic enzyme gene expression via different pathways. In addition, authors elucidated the transcriptional and post-transcriptional regulatory mechanisms of several TFs involved in alkaloid biosynthesis. Finally, they discussed the use of TFs as powerful tools for alkaloid engineering.

Major:

  1. Evolutionary aspects of TFs should be discussed more and depicted through the figure.
  2. It will be better if authors can analyse the correlation between different TFs involved in alkaloid biosynthesis via different pathways.
  3. The application of TFs gene clusters in alkaloid biosynthesis is not very vibrant in the article, as mentioned in the abstract.

Minor:

  1. Line 62: Full form of DREB should be written for the first time.
  2. Line 170: Spelling mistake - AtMYC2 instead of AtMCY2.
  3. Line 198: Heading - Genes instead of gene.
  4. Line 336: Lack of reference.
  5. Please check this line: The DREB and ERF subfamily proteins 64 are known to bind to the GCC box (e.g., AGCCGCC) and the dehydration-responsive element (e.g., GCCGAC), respectively. Usually, ERF binds with GCC box.

Author Response

(The authors gave the same response as above.)

Round 2

Reviewer 3 Report

1. As the authors did not discuss in detail about evolutionary perspectives of TFs, they have to rephrase the lines of the abstract: "we discuss the evolution of transcription factors as gene clusters in alkaloid biosynthesis and their application in alkaloid engineering" .

2. Authors can easily generate an interaction network by converting all other plants TFs sequences to Arabidopsis sequences for analysis.
